# Anticipated Capabilities of the ODYSEA Wind and Current Mission Concept to Estimate Wind Work at the Air–Sea Interface

Hector Torres [1,*], Alexander Wineteer [1], Patrice Klein [2,3], Tong Lee [1], Jinbo Wang [1], Ernesto Rodriguez [1], Dimitris Menemenlis [1] and Hong Zhang [1]

1   Jet Propulsion Laboratory, California Institute of Technology, Pasadena, CA 91109, USA; alexander.g.wineteer@jpl.nasa.gov (A.W.); tlee@jpl.nasa.gov (T.L.); jinbo.wang@jpl.nasa.gov (J.W.); ernesto.rodriguez@jpl.nasa.gov (E.R.); dimitris.menemenlis@jpl.nasa.gov (D.M.); hong.zhang@jpl.nasa.gov (H.Z.)
2   Environmental Science and Engineering, California Institute of Technology, Pasadena, CA 91125, USA; pklein@caltech.edu
3   LMD/IPSL, CNRS, Ecole Normale Supérieure, PSL Research University, 75005 Paris, France
*   Correspondence: hector.torres.gutierrez@jpl.nasa.gov

**Abstract:** The kinetic energy transfer between the atmosphere and oceans, called wind work, affects ocean dynamics, including near-inertial oscillations and internal gravity waves, mesoscale eddies, and large-scale zonal jets. For the most part, the recent numerical estimates of global wind work amplitude are almost five times larger than those reported 10 years ago. This large increase is explained by the impact of the broad range of spatial and temporal scales covered by winds and currents, the smallest of which has only recently been uncovered by increasingly high-resolution modeling efforts. However, existing satellite observations do not fully sample this broad range of scales. The present study assesses the capabilities of ODYSEA, a conceptual satellite mission to estimate the amplitude of wind work in the global ocean. To this end, we use an ODYSEA measurement simulator fed by the outputs of a km scale coupled ocean–atmosphere model to estimate wind work globally. The results indicate that compared with numerical truth estimates, the ODYSEA instrument performs well globally, except for latitudes north of 40°N during summer due to unresolved storm evolution. This performance is explained by the wide-swath properties of ODYSEA (a 1700 km wide swath with 5 km posting for winds and surface currents), its twice-a-day (daily) coverage at mid-latitudes (low latitudes), and the insensitivity of the wind work to uncorrelated errors in the estimated wind and current.

**Keywords:** Doppler scatterometer; winds; surface currents; wind work

## 1. Introduction

Wind work at the air–sea interface is the transfer of kinetic energy between the ocean and the atmosphere, which is a fundamental driver of ocean circulation [1]. Recent estimates from global coupled ocean–atmosphere models run with km scale resolution point to total wind work of a magnitude up to ∼5 terawatts over the whole ocean, i.e., five times larger than reported ten years ago [2,3]. Approximately ∼28% of this wind work is used to generate near-inertial oscillations and internal gravity waves (with spatial scales of 20 to 1000 km and time scales of hours), which impact ocean mixing and therefore contribute to setting global stratification and large-scale circulation [3–5]. Another ∼28% contributes to force and damp mid-latitude currents, such as those associated with mesoscale eddies (with spatial scales from 50 km to more than 500 km and time scales of days to months), which are critical players in the horizontal and vertical transport of heat at these latitudes [3,6,7]. The remaining ∼44% forces large-scale zonal jets with time scales of a few months, particularly

in equatorial and tropical latitudes, where they play a pivotal role in the El Niño–Southern Oscillation (ENSO) [3,8].

Wind work ($F_s$) is defined in this study as a function of wind vectors observed at a height of 10 m ($\mathbf{U_{10}}$) and surface current vectors ($\mathbf{u_o}$) [2,9]:

$$F_s = \rho_{air} C_d |\mathbf{U_{10}} - \mathbf{u_o}|(\mathbf{U_{10}} - \mathbf{u_o}) \cdot \mathbf{u_o}. \tag{1}$$

where $\rho_{air}$ is the air density and $C_d$ a drag coefficient [10]. The large amplitude of the total wind work is explained by the multiscale characteristics of winds and surface currents that need to be taken into account: scales from one hour to at least one year, and 10 km to more than 3000 km. In addition, wind work is sensitive to the collocation and contemporaneity of winds and surface currents [3,9]. For example, a phase shift of 12 h between winds and surface currents can reduce the forcing of near-inertial oscillations and internal gravity waves by a factor up to 5–10 [3]. Similarly, taking into account the collocation in space of wind anomalies and ocean mesoscale eddies leads to reduced damping of ocean eddies by the atmosphere [6,9]. Because of their importance to ocean circulation, these numerical results have to be confirmed by observations, which require satellite observations of winds and surface currents that are collocated and contemporaneous with a spatial resolution of about 10 km and a temporal resolution of less than 12 h [3]. Such requirements are not met by the present satellite observations.

The National Academies' 2018 Decadal Survey recommended a new competed Explorer class mission for surface current and wind measurements, with a spatial resolution of 5–10 km, a temporal resolution of 12 h, a global coverage time of 1–2 days, and a random error of less than 50 cm/s for surface currents and 1 m/s for winds [11]. Rodríguez et al. [12] and Wineteer et al. [13] showed that these goals can be met by a new Doppler scatterometer concept. The resulting conceptual satellite mission is called Ocean Dynamics and Sea Exchanges with the Atmosphere (ODYSEA). The present study assesses the potential capabilities of ODYSEA to simultaneously measure winds and surface currents over the global oceans in order to diagnose wind work. To this end, we make use of outputs of winds and surface currents from a km scale coupled ocean–atmosphere simulation [3,14]. These outputs are used to feed a Doppler scatterometer instrument and observation simulator, hereinafter called the ODYSEA simulator [13]. The ODYSEA simulator is presented in the next section. The results are shown and discussed in Section 3. The conclusions follow in the last section.

## 2. The ODYSEA Simulator

Rodríguez et al. [12] described a Doppler scatterometer mission design fitting within the NASA Earth Explorer class mission specifications that satisfies Decadal Survey requirements. This ideal design utilizes a 5.0 × 0.35 m antenna, 400 W transmit power, a 650 km sun-synchronous polar orbit with a 4-day exact repeat cycle, and an incidence angle of about 56°. This results in a wide measurement swath of about 1700 km, global coverage better than 90% of the ocean daily, and a spatial posting of 5 km. More recent design exercises have resulted in a decreased antenna length to fit with the NASA Earth System Explorer cost cap and launch vehicle requirements, resulting in an increased surface current measurement error relative to the 5 m antenna case described in Rodríguez et al. [12]. Figure 1 shows an example of the sampling expected from ODYSEA.

For this study, we use the most recent version of the ODYSEA simulator, which takes into account updated instrument performance and mission design. The simulator generates ODYSEA science data sampling using outputs from a coupled ocean–atmosphere simulation (hereafter called COAS) run on a global scale at high resolution (see Section 2.2). From there, surface currents and wind speed errors estimated according to Rodríguez et al. [12] and Wineteer et al. [13] are added onto the collocated winds and surface currents from COAS.

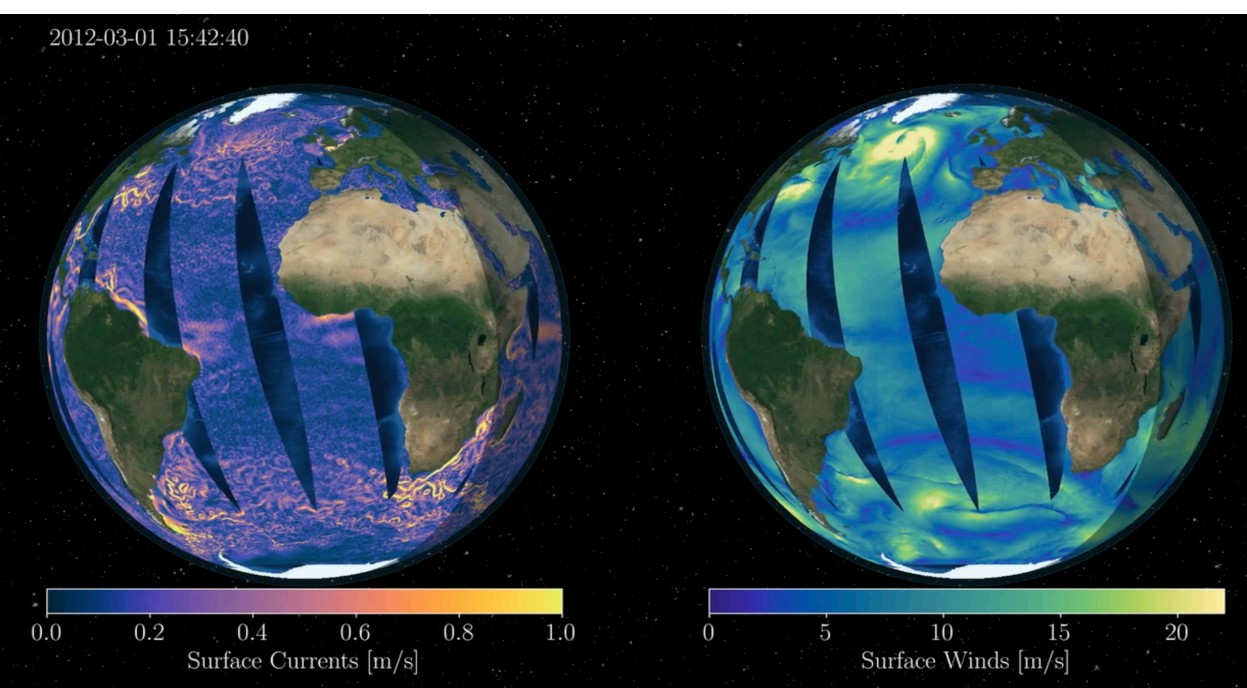

**Figure 1.** Sampling of surface ocean currents (**left pane**l) and ocean winds (**right panel**) by the ODYSEA simulator.

## 2.1. Surface Current Errors

Baseline ODYSEA performance is often cited as 50 cm/s at 5 km resolution, an uncertainty value that is typical given the most likely ocean wind speed and averaged across the swath. In reality, surface current measurement errors from ODYSEA vary strongly depending on the wind speed and look geometry (Wineteer et al. [13]). The center and edges of the measurement swath have increased errors due to poor azimuth diversity that make vector inversion difficult, and the resulting uncertainty can reach values much beyond the baseline value in those regions (see Figure 2 in Wineteer et al. [13]). Since the radar signal-to-noise ratio depends on the amount of backscattered radar power, higher wind speeds that roughen the ocean surface will increase the radar signal-to-noise-ratio and decrease surface current uncertainty, as shown in Rodríguez et al. [12] and Wineteer et al. [13]. Figure 2 in Wineteer et al. [13] shows that the surface current uncertainty at a normalized distance of 0.5 from the center of the swath can vary from 0.05 m/s for high winds (12 m/s) to >1 m/s for very low wind speeds (<3 m/s). Figure 2d shows a scatterplot of surface current uncertainty across wind speed that corresponds to spatial fields shown in Figure 2a. The largest errors on the extreme edges of the track have been removed, primarily leaving variability due to wind speed. In regions of very low winds, such as in the tropics, ODYSEA surface current errors grow exponentially. Conversely, in regions of >6 m/s wind speeds, which represents 75% of wind events in 3 months (Figure 2c), performance is significantly better than the baseline requirement.

## 2.2. The Coupled Ocean–Atmosphere Simulation (COAS)

The coupled ocean–atmosphere simulation is based on the Goddard Earth Observing System (GEOS) atmospheric and land model coupled with the ocean component of the Massachusetts Institute of Technology general climate model (MITgcm). The COAS configuration used in this study is fully described in Strobach et al. [14] and Torres et al. [3]. The atmospheric model has a nominal horizontal grid spacing of $\sim 1/16°$ ($\sim$6 km ) and 72 vertical levels, and the ocean model has a nominal horizontal grid spacing of $1/24°$ ($\sim$4 km ) and 90 vertical levels. In terms of physical scales, the spacing grid resolution of 6 km and 4 km are sufficient to resolve length scales between 25 km and 30 km [15]. The COAS simulation was initialized on 20 January 2012, using initial ocean conditions from

the forced LLC2160 MITgcm simulation and 2020 atmospheric initial conditions from the Modern-Era Retrospective analysis for Research and Applications, Version 2 (MERRA-2) interpolated to the GEOS grid [3]. The numerical simulation cannot resolve the mechanical boundary layers that must be present at the air–sea interface and seafloor, resulting in the transfer of momentum. The numerical machinery uses a drag stress at the ocean top and bottom that scales quadratically with the near-boundary velocity [16].

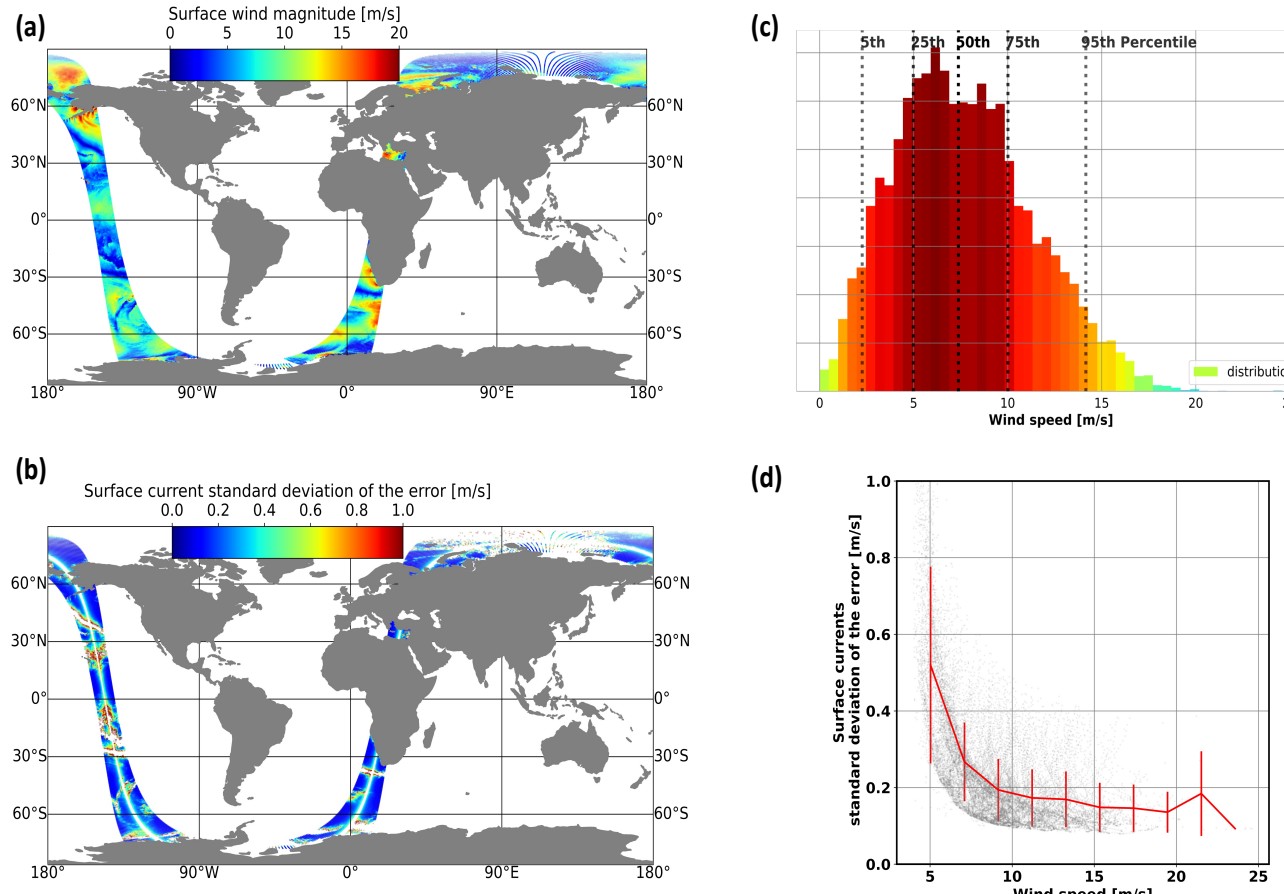

**Figure 2.** Ocean current standard deviation of the error dependence on wind speed along a track in January. (**a**) Surface wind magnitude. (**b**) Surface current standard deviation of the error geographically oriented. (**c**) Histogram of wind speed events for 3 months: January–February–March. (**d**) Surface current standard deviation of the error as a function of the wind speed from the panels (**a,b**). These plots were generated using the ODYSEA simulator and COAS outputs.

Surface currents involve much smaller scales than winds and evolve more slowly. This is illustrated in Figure 1, which shows surface currents (left panel) and winds (right panel) sampled by the ODYSEA simulator in the global ocean. The wind field involves large scales $\mathcal{O}$ (1000 km), resulting from atmospheric weather patterns that propagate rapidly. Embedded within these large-scale patterns are smaller-scale patterns (as small as 100 km), some of them propagating with the large-scale ones. Energetic surface currents include near-inertial motions and internal tides, mesoscale eddies, and zonal jets.

*2.3. Synthetic Datasets*

The ODYSEA simulator estimates wind and current errors and adds them onto collocated winds and currents from COAS. The simulator uses a simple model for the wind errors to simulate ODYSEA performance requirement: the error standard deviation corresponds to 10% of the wind speed when the speed is larger than 10 m/s and 1 m/s for smaller wind speeds. For surface currents, an error model based on [12] was used to form

a lookup table based on wind speed, wind direction, and swath position. At the center and edges of the swath, the resulting surface current errors are large. In this study, we exclude the center 100 km and 50 km on either edge of the swath in addition to areas with winds less than 5 m/s, because the standard deviation of the error is larger than the scientific requirements. To separate the effects of measurement noise from sampling, two experiments were run. The first, ODYSEA-NF (noise-free), uses only ODYSEA sampling (including center, edge, and wind speed blanking) but no measurement noise. The second, ODYSEA-N (noisy), adds measurement noise in addition to the sampling in ODYSEA-NF.

Figure 3a shows an example of surface current sampling from ODYSEA-NF in the Tropical Pacific. White areas identify blanked regions where surface current uncertainty becomes unacceptable. Figure 3b shows surface currents as sampled by ODYSEA in addition to estimated measurement noise (ODYSEA-N). Compared with Figure 3a, measurement noise is noticable, particularly in regions with weak surface current and low wind speed, a combination in which the signal-to-noise ratio is dominated by the random noise. The resulting ODYSEA-NF and ODYSEA-N datasets will give us insights into the potential effects of the sampling on the wind work by the ODYSEA instrument and whether the random noise has an impact on the estimations of the wind work.

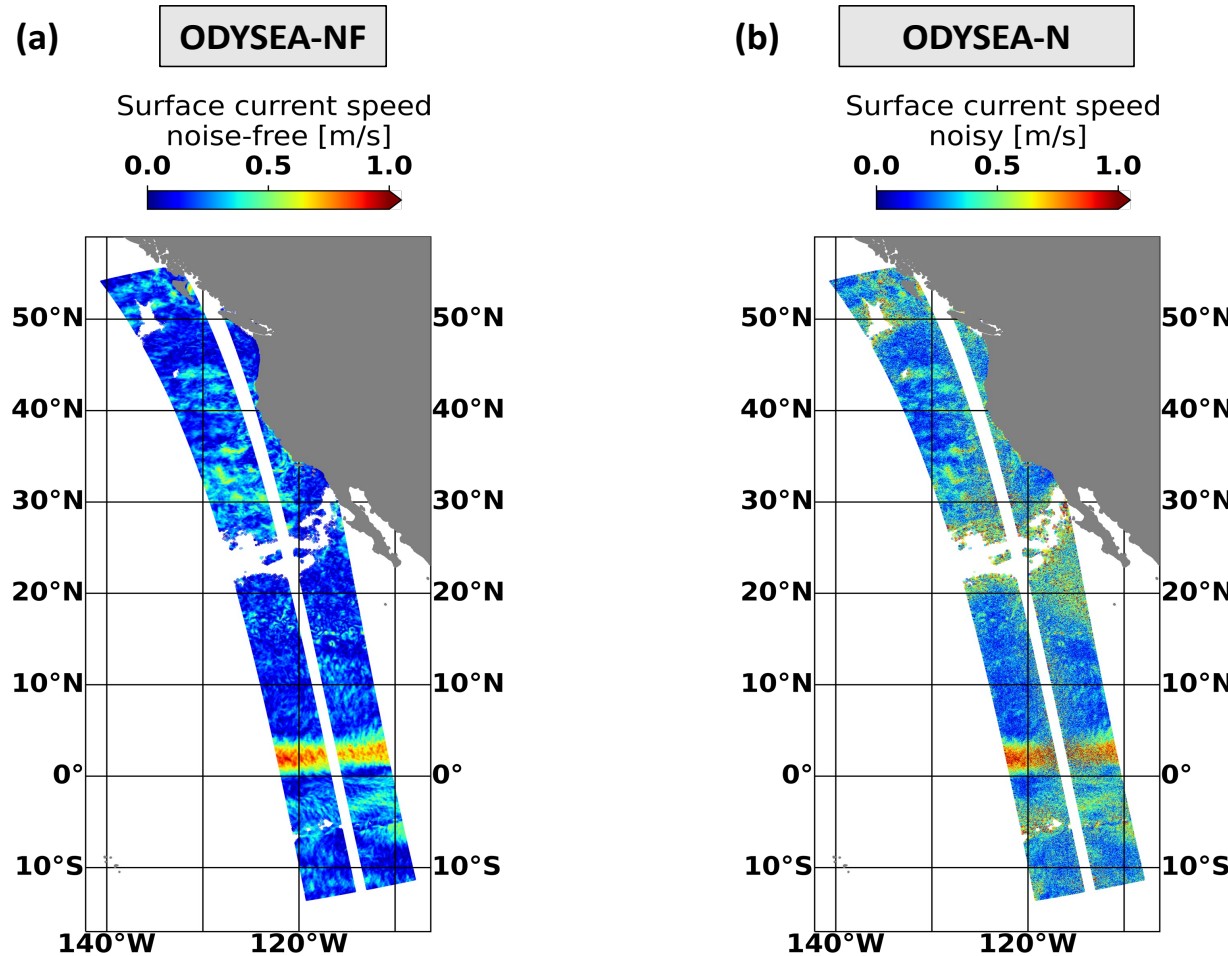

**Figure 3.** Surface current speed for ODYSEA simulated passes over the eastern Pacific Ocean at 5 km resolution. The left panel (**a**) shows the surface current speed without error but removing data where the wind speed is smaller than 5 m/s. The right panel (**b**) shows the surface current speed plus the current errors due to the wind speed. Since the wind speed is large-scale (>500 km), the current errors are large-scale.

Figure 4a–c show the global coverage of ODYSEA after 1.5 days with the center
and edges of the swaths removed. Within this short time period, ODYSEA is able to
sample more than 90% of the global ocean with a spatial resolution of 5 km. Figure 4a
reveals a smooth transition of surface currents between swaths: for latitudes larger than
10°N and 10°S, slowly evolving surface currents ($u_o$) such as large-scale currents and
mesoscale eddies are well sampled by the temporal resolution of ODYSEA (12 h for these
latitudes). In contrast, the wind stress field, $U_{10}$ (Figure 4b), at mid- and high latitudes
displays some discontinuities in particular at southern and northern mid-latitudes (for
example at 140°W, 48°S and 150°W, 48°N) where atmospheric storm tracks are located.
Such discontinuities are due to the ODYSEA 12 h temporal sampling that cannot fully
resolve the fast propagation of atmospheric storms. The resulting wind work field obtained
from Equation (1) (Figure 4c) does not show clear discontinuities such as in the wind field.
Large positive values at mid-latitudes are well captured, being localized where large-scale
atmospheric patterns are located (see Torres et al. [3]). Tropical and equatorial latitudes
(smaller than 10°N and 10°S) are under-sampled in a few regions (as revealed by the grey
areas in Figure 4), since the temporal resolution is larger than 12 h at these low latitudes.

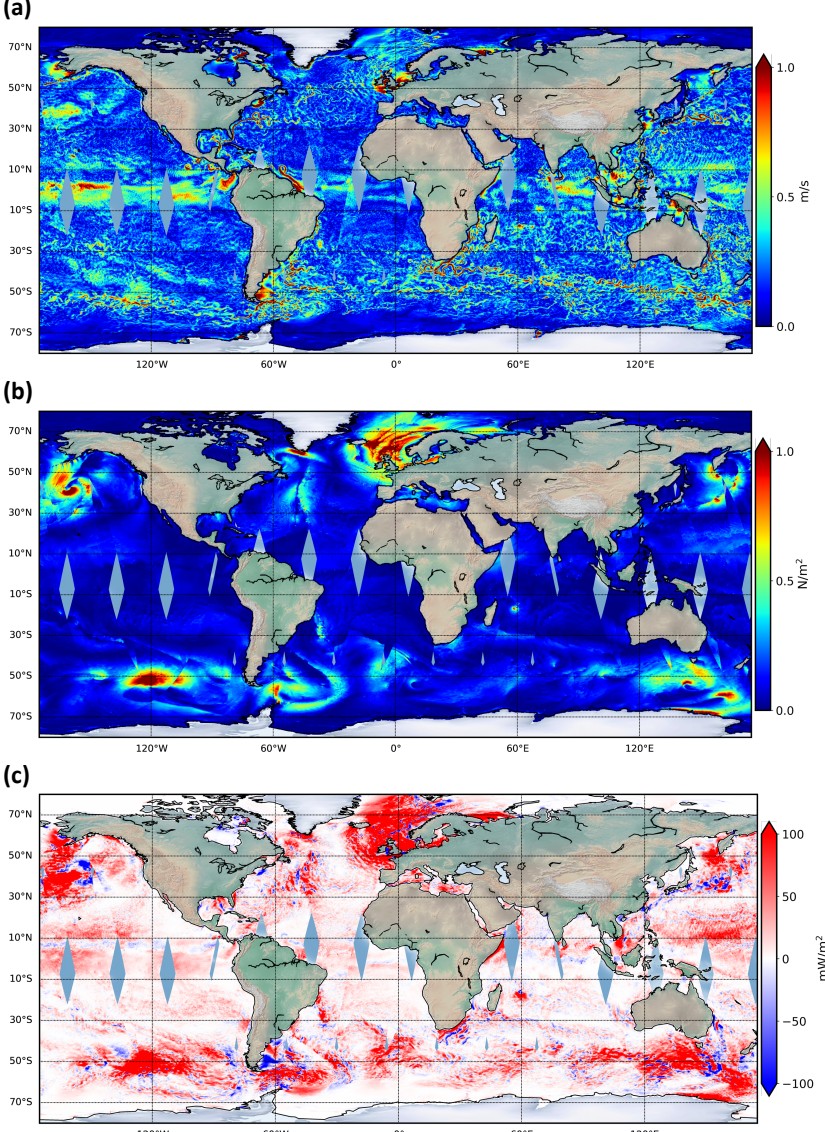

**Figure 4.** Global coverage of ODYSEA-NF simulated data: (**a**) surface ocean currents ($u_o$), (**b**) surface
wind stress ($F_s$), and (**c**) wind work (using Equation (1)).

In addition to the ODYSEA-NF and ODYSEA-N datasets, we have generated two other datasets to mimic the existing satellite products (like AVISO, Globcurrents, and QuickSCAT). The first, called "AVISO + QuickScat-like", implements a spatial smoothing of 130 km for surface currents and 50 km for winds, and a temporal average of 7 days for surface currents and 12 h snapshots for winds. This dataset is labeled as $U_{130km}^{7d} + \tau_{50km}^{12h}$. The second one, called "Globcurrent + QuickScat-like", implements similar spatial smoothing but a temporal average of 1 day for surface currents and 12 h snapshots for winds and is called $U_{130km}^{1d} + \tau_{50km}^{12h}$. Table 1 summarizes all the datasets used in this study.

**Table 1.** Dataset of ocean currents and ocean winds at given spatial and temporal resolution.

| Data | Spatial Resolution | Temporal Resolution |
|---|---|---|
| COAS | 4 km | 1 h |
| ODYSEA-NF (noise-free) | 5 km | 12 h |
| ODYSEA-N (noisy) | 5 km | 12 h |
| $U_{130km}^{7d} + \tau_{50km}^{12h}$ ("AVISO + QuickScat-like") | $U$ at for 130 km and $\tau$ at 50 km | $U$ at for 7-day and $\tau$ at 12 h |
| $U_{130km}^{1d} + \tau_{50km}^{12h}$ ("Globcurrent + QuickScat-like") | $U$ at for 130 km and $\tau$ at 50 km | $U$ at for 1-day and $\tau$ at 12 h |

## 3. Results

We now assess the capabilities of the ODYSEA instrument to diagnose the wind work at the air–sea interface by comparing data from COAS with the ODYSEA-NF and ODYSEA-N datasets generated by the ODYSEA simulator. This comparison makes use of the seasonal wind work fields (averaged over 3 months) analyzed during two seasons: January–February–March (or Jan-Feb-Mar) and July–August–September (or Jul-Aug-Sep).

### 3.1. Seasonal and Geographical Variability of the Total Wind Work

Figure 5 shows the spatial distribution of wind work over the global oceans. The left panels represent wind work from the full COAS model fields, and the right panels from using the ODYSEA-N sampling and noise. The upper panels are for the Jan-Feb-Mar season and the bottom panels for the Jul-Aug-Sep season. Positive wind work means an injection of kinetic energy from the atmosphere to the ocean with the opposite being true for negative values.

The global estimates of the total wind work using ODYSEA-N are consistent with those using COAS. The panels in Figure 5 reveal a strong similarity between COAS and ODYSEA-N wind work in terms of their spatial distribution and magnitudes. As discussed in Torres et al. [3], significant seasonality is observed at mid-latitudes in each hemisphere, with the wind work intensified in Jan-Feb-Mar in the northern hemisphere and Jul-Aug-Sep in the southern hemisphere. This intensification is due to the mid-latitude atmospheric storm tracks being more energetic in winter than in summer. Wind work at tropical and equatorial latitudes, i.e., between 30°S and 30°N, display zonal patterns with a weaker seasonal variation. Such zonal elongated patterns are known to be associated with westward trade winds [17]. South of Japan, some meridional streaks emerge in Jul-Aug-Sep, which are associated with atmospheric tropical cyclones [3]. Although these cyclones propagate quite fast, they are well captured by the ODYSEA simulator.

The overall agreement between COAS and ODYSEA-N datasets is confirmed in the scatter plots shown in the inset of Figure 5. This agreement is remarkable in Jan-Feb-Mar. In Jul-Aug-Sep, a slight deviation from the main diagonal is observed for values smaller than 40 mW/m² : ODYSEA-N wind work appears to be underestimated (with respect to COAS) for these values. This discrepancy is better revealed by the latitudinal profiles of the wind work zonally integrated (Figure 6): although no major differences are found between COAS and ODYSEA-N in Jan-Feb-Mar (respectively green and orange curves in Figure 6a),

COAS and ODYSEA-N wind work differ in the northern hemisphere (north of 40°N) in Jul-Aug-Sep ( Figure 6b), with this difference being ∼50% of the total magnitude.

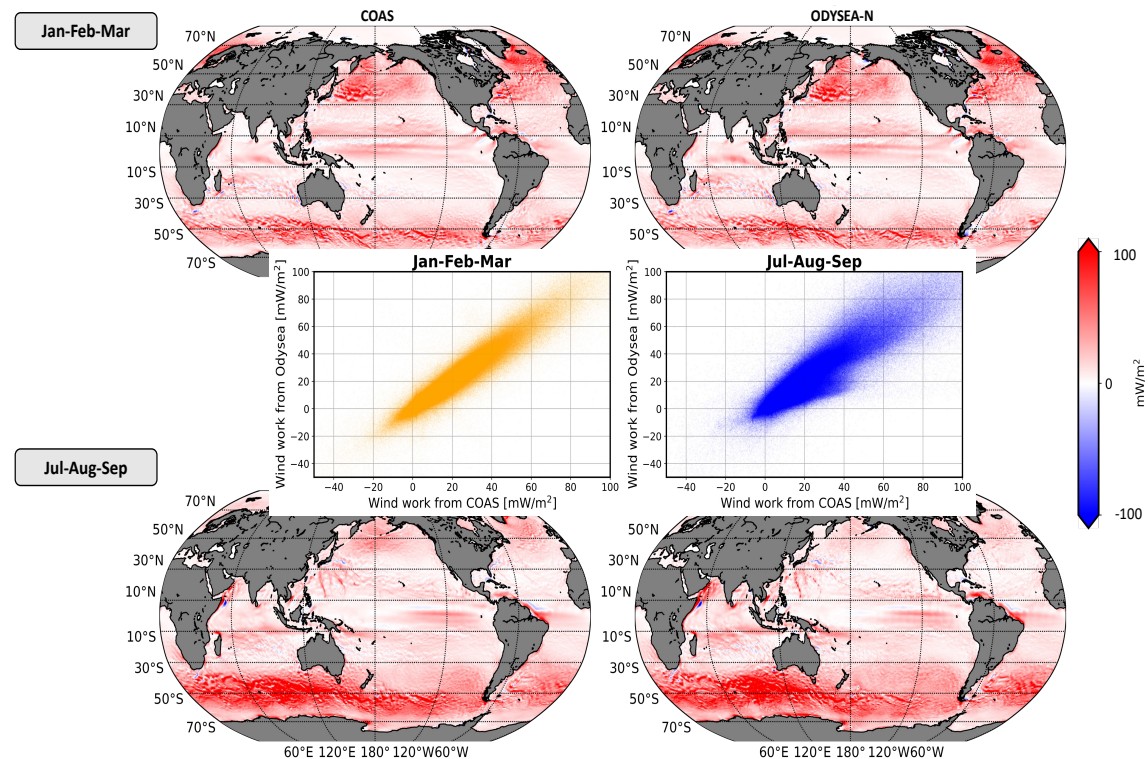

**Figure 5.** Seasonal average of wind work from COAS (**left column**) and from ODYSEA-N (**right column**) during Jan-Feb-Mar (**first row**) and Jul-Aug-Sep (**second row**). The inset shows a scatter plot of wind work between COAS (x-axis) and ODYSEA-N (y-axis).

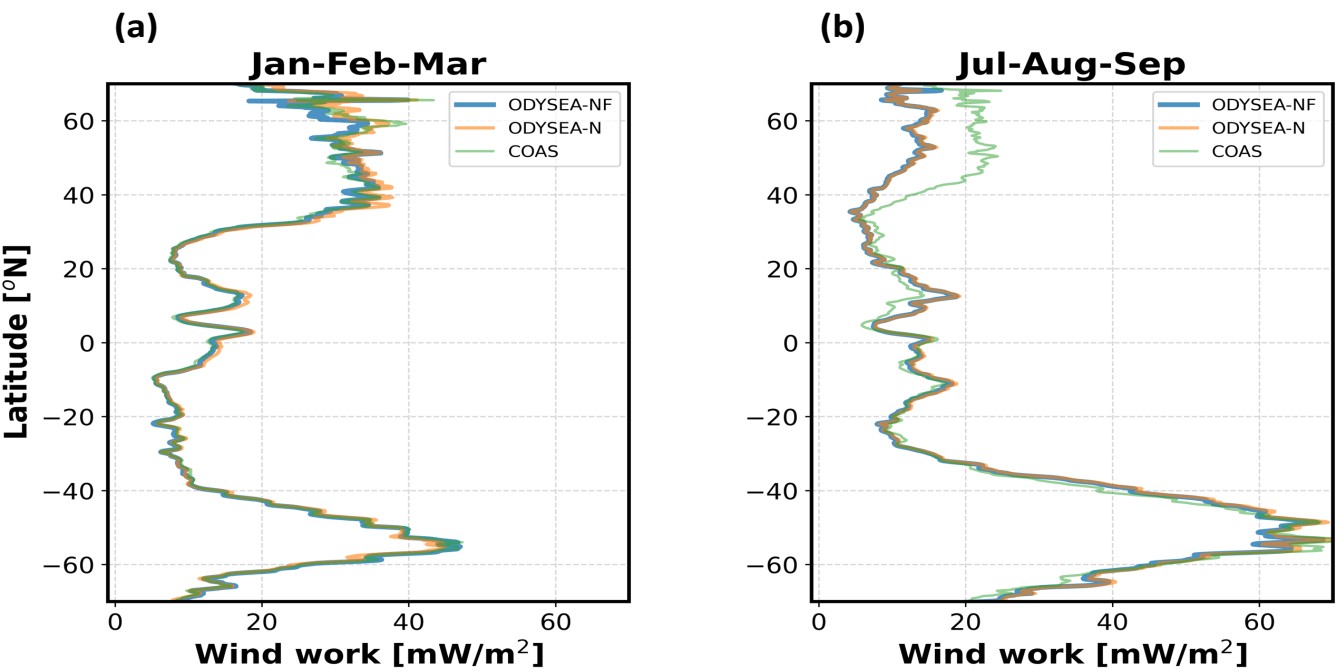

**Figure 6.** (**a**,**b**) Latitudinal profile of wind work zonally integrated then divided by the ocean area in the latitude band. ODYSEA-NF data (blue line), ODYSEA-N (orange line), and COAS (green line).

To understand whether the significant difference observed in Figure 6b is due to surface current errors or sampling, we compare the wind work from ODYSEA-N (no surface current random errors, only sampling). No differences are observed between ODYSEA-NF (blue curve on Figure 6) and ODYSEA-N (orange curve) either in Jan-Feb-Mar or Jul-Aug-Sep. Figure 7 shows the differences between COAS and ODYSEA-N when different random errors for surface currents are used in ODYSEA-N, respectively, a baseline of 0.35 m/s (blue curve) and 1 m/s (orange curve). The similarity between these two curves further confirms that the difference between COAS and ODYSEA-N found in the northern hemisphere in Jul-Aug-Sep is not due to random measurement errors. ODYSEA sampling is the same in winter and summer, and winds and surface currents have spatial and temporal scales that do not change much between these two seasons. So, the good results obtained in Jan-Feb-Mar in the northern hemisphere exclude sampling errors as an explanation for the difference found in summer. The only explanation we have so far is that this difference may be due to the high-frequency ($>f$) low wind amplitudes during summer in this region that force near-inertial oscillations. These low winds are not taken into account in the ODYSEA simulator when their amplitude is less than 5 m/s. This does not occur in the summer southern hemisphere, since wind amplitudes are usually larger than 5 m/s. This explanation needs to be confirmed.

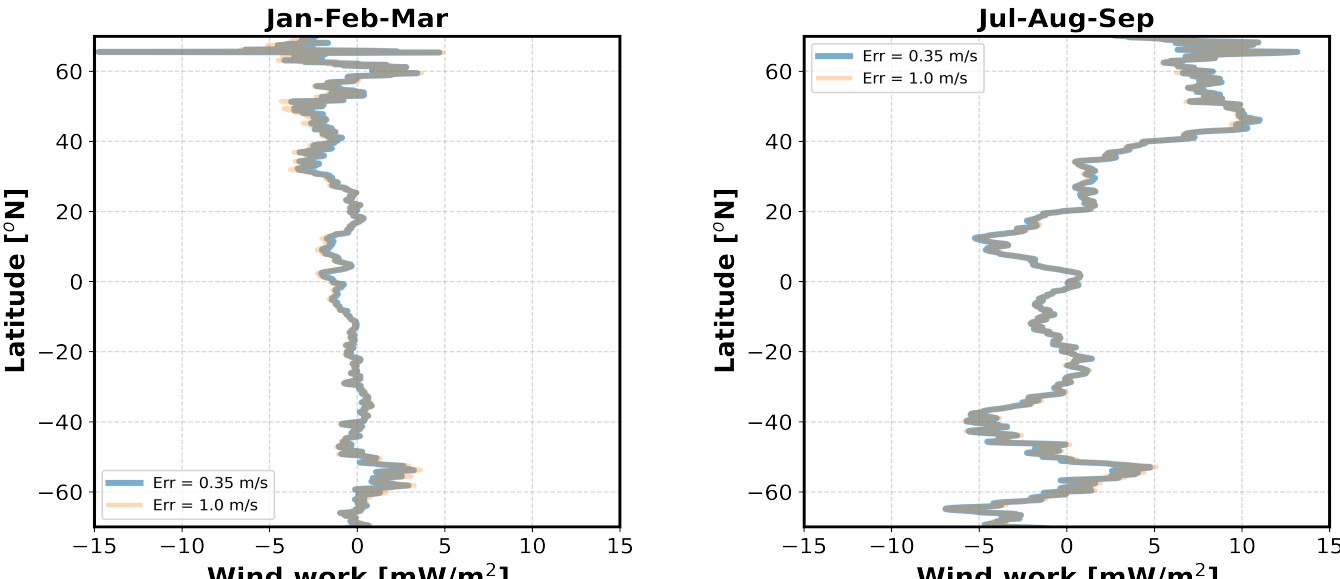

**Figure 7.** Latitudinal profiles of the differences between the wind work from COAS and the wind work generated by the ODYSEA simulator with different baseline errors for surface currents. The blue curve corresponds to the difference between COAS and ODYSEA-N (baseline current error of 0.35 m/s). The orange curve corresponds to the difference between COAS and a worse scenario where the baseline current error is 1 m/s. These profiles correspond to the wind work zonally integrated and then divided by the ocean area in the latitude band.

### 3.2. Mesoscale Wind Work

As mentioned before, the total wind work considered so far include several components such as (i) the high-frequency component ($>1/3$-day), involving winds and currents with a time scale smaller than 1–3 days, known to force near-inertial waves; (ii) the low-frequency components ($<1/3$-day) that force or damp large-scale motions and ocean mesoscale eddies; as well as (iii) the seasonal component that impacts large-scale zonal jets, in particular those at the equator. In the context of the present study, we have focused on the performance of ODYSEA to diagnose the wind work component associated with ocean mesoscale eddies, since this component is known to damp mesoscale eddies that explain almost 80% of the total kinetic energy in the oceans. It is also the smallest component

of the wind work and may therefore be sensitive to ODYSEA instrumental errors. The methodology to diagnose this specific component is fully described in Torres et al. [3] and briefly outlined in Appendix A. This component, which corresponds to the last term in Equation (A2), is referred to here as the mesoscale wind work component.

Figure 8 shows the mesoscale wind work component estimated from COAS (panel a) and estimated using the ODYSEA-N dataset (panel b). This wind work is mostly negative, corresponding to the damping of mesoscale eddies, and has a net magnitude 1/(10–20) smaller than the total wind work. The physics of this damping of mesoscale eddies is discussed in [3] (see also [18]). Note that this physics includes the Ekman pumping [19]), since the latter is implicitly included in the full momentum equations resolved by the coupled model. It is principally found at mid-latitudes, where mesoscale eddies are energetic, particularly in western boundary currents (Gulf Stream and the Kuroshio) and the Antarctic Circumpolar Current. This component is slightly positive in isolated regions, mostly located in the tropical and equatorial bands, and is associated with tropical wave instabilities [3]. The difference between COAS and ODYSEA-N wind work is shown in panel c in Figure 8. Quantification of this difference is shown in Figure 9. As expected from Figure 8, the mesoscale wind work component is larger at mid-latitudes than in tropical and equatorial regions (Figure 9a). The differences between COAS and ODYSEA-N (Figure 9b), however, are less than 10% of the magnitude of this component. These results emphasize the performance of the ODYSEA instrument to estimate this component of the wind work known to be important for ocean dynamics [6,9].

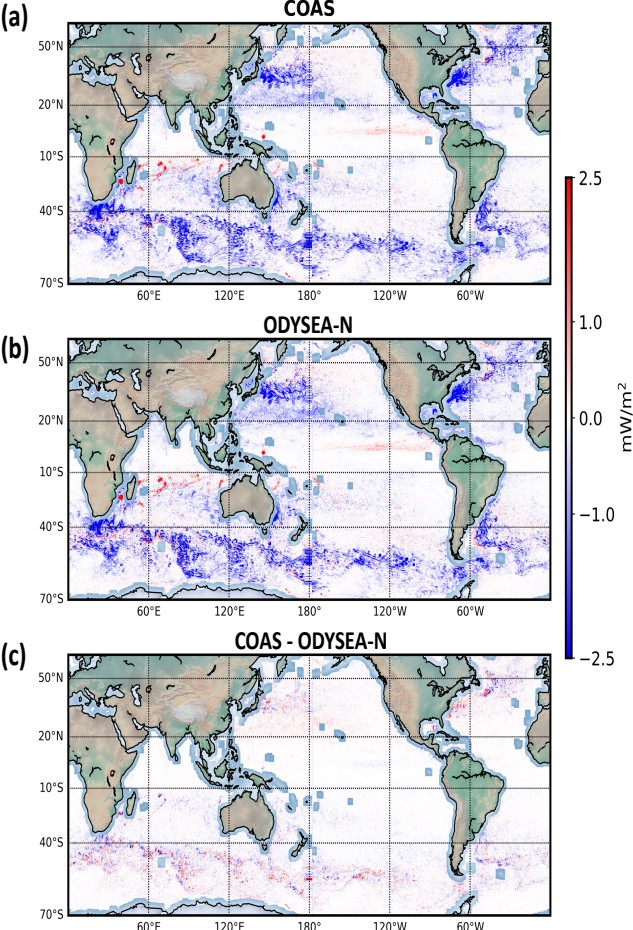

**Figure 8.** Low-frequency wind work with scales smaller than *Lc* averaged over Jan-Feb-Mar, $\overline{\tau'_{lf<}.\mathbf{u_o}'_{lf<}}$. (**a**) Low-frequency wind work estimated from COAS, (**b**) low-frequency wind work estimated from ODYSEA-N, and (**c**) difference between panel COAS and ODYSEA-N.

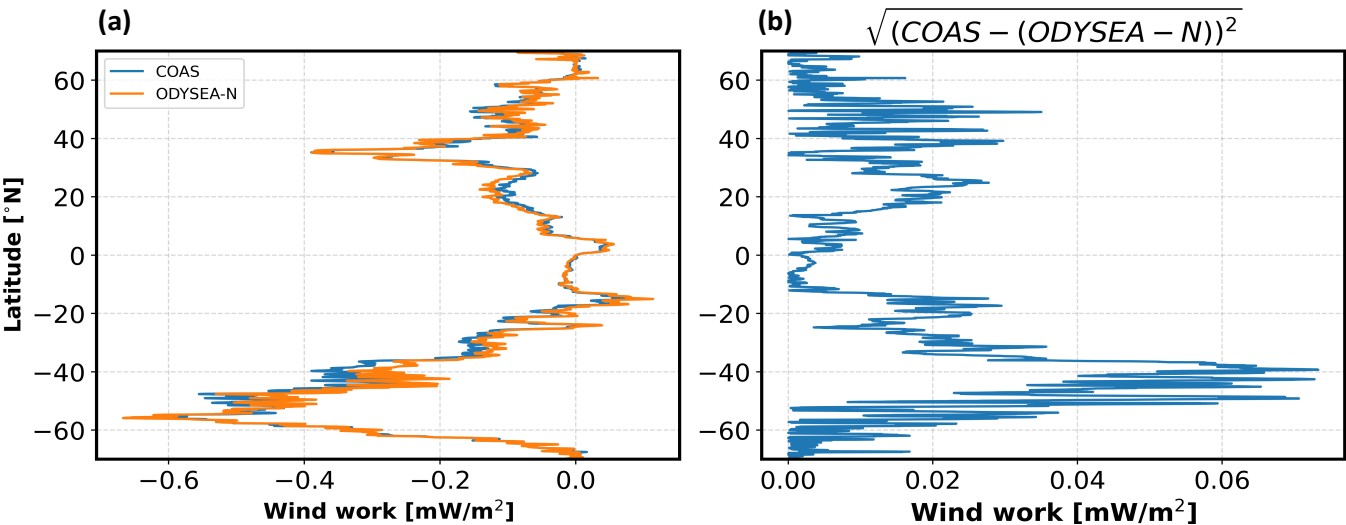

**Figure 9.** (**a**) Low-frequency (<1/3-day) wind work from Figure 8 multiplied by the area of the numerical grid cell (m²) and zonally integrated, then divided by the ocean area in latitude band. (**b**) Difference between COAS and ODYSEA-N.

### 3.3. Comparison with Wind Work from Existing Satellite Observations

A last question is: What are the improvements expected from ODYSEA with respect to existing satellite sensors such as QuickSCAT for winds and conventional altimeters for geostrophic currents? To answer this question, we refer to the "AVISO + QuickScat-like" and "Globcurrent + QuickScat-like" datasets defined in Section 2.3 (see Table 1). The wind work diagnosed from these datasets (red and purple curves in Figure 10, respectively for "AVISO + QuickScat-like" and "Globcurrent + QuickScat-like") is substantially reduced in magnitude compared with the wind work estimated in ODYSEA-N (orange curve). The "AVISO + QuickScat-like" and "Globcurrent + QuickScat-like" datasets do not include high-frequency motions such as near-inertial motions, which means that the wind work component that forces near-inertial motions is missing when using existing satellite observational studies of wind work. These results confirm the need to measure winds and currents over a broad range of scales.

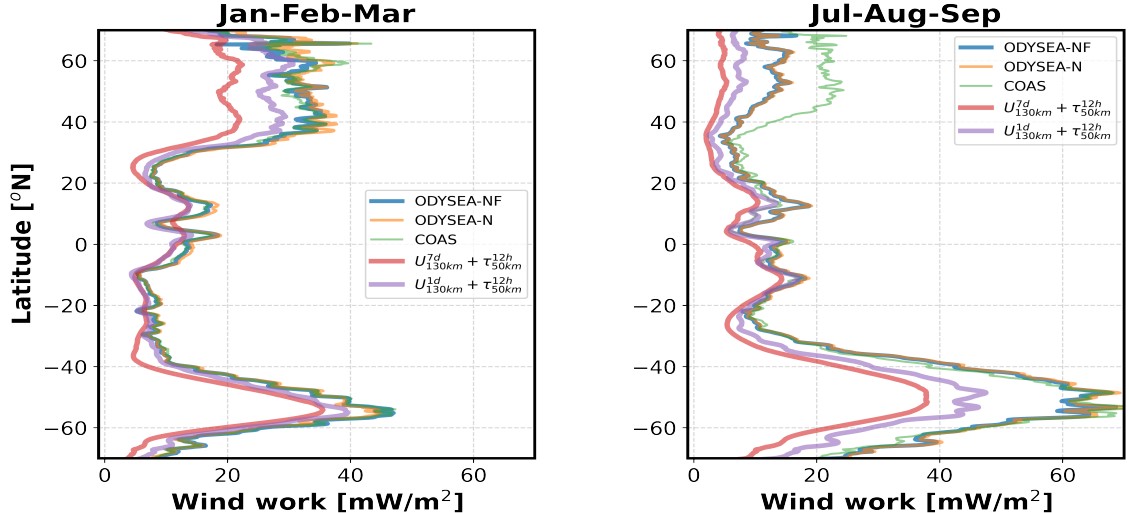

**Figure 10.** Latitudinal profile of wind work zonally integrated. Similar to Figure 6 but with the additional profile of wind work estimated from "AVISO + QuickScat-like" ($U^{7d}_{130km} + \tau^{12h}_{50km}$) (red line) and from "Globcurrent + QuickScat-like" ($U^{1d}_{130km} + \tau^{12h}_{50km}$) (purple line).

Another benefit of ODYSEA is the ability to take into account the collocation and contemporaneity of winds and currents, as mentioned before, which is not the case with the existing satellite observations. These collocation and contemporaneity properties are critical not only for the wind work that forces near-inertial motions [3] but also for the wind work that damps mesoscale eddies (for details of the mesoscale wind work, see Section 3.2) as revealed by the results presented in Appendix B. These results indicate that the mesoscale wind work estimated using an uncoupled ocean-only simulation (LLC), with an ocean configuration identical to COAS but forced by ECMWF winds, is twice as large as the one from COAS. This difference is explained by the impact of mesoscale ocean eddies and their associated SST on the local wind field [14,20–25]. COAS takes into account this impact, but not the ocean-only simulation: ECMWF winds are not affected by mesoscale eddies from the ocean-only simulation, since these winds do not "know" the mesoscale eddies from COAS. The existing satellite observations poorly take into account this collocation and contemporaneity (QuickScat sampled once daily and altimeter data every 10 days), as revealed by the results from Rai et al. [26]: the mesoscale wind work estimated from these observations is almost twice as large as the one estimated from COAS but close to the one diagnosed from the ocean-only simulation (see Figure A1).

## 4. Conclusions

The wind work considered in this study is one of the major drivers of ocean dynamics, which impacts atmospheric weather and climate change [24,25]. The large magnitude of the wind work, recently revealed by high-resolution coupled numerical simulations [2,3], is explained not only by the contribution of the broad range of spatial and temporal scales covered by winds and surface currents but also by the strong sensitivity of the wind work to the collocation and contemporaneity of winds and currents at all spatial and temporal scales [14,27,28]. The existing satellite observations do not take into account these wind work characteristics and, therefore, cannot confirm results from high-resolution coupled simulations. The present study has explored the future capabilities of a new satellite mission, ODYSEA, to diagnose the wind work globally. To this end, we have used numerical outputs of winds and surface currents from a km scale coupled ocean–atmosphere (COAS) to feed a purposely designed ODYSEA simulator.

Comparisons of the wind work between ODYSEA-like data and COAS are consistent with a difference of ∼0.5 mW/m$^2$ when seasonally and zonally averaged over 3 months. ODYSEA sampling is sufficient to capture the currents and winds that primarily determine wind work. This comparison still holds even if the random error of surface currents increases to 1 m/s. An additional exercise was conducted to quantify the impact of uncorrelated ODYSEA measurement errors on the mesoscale wind work. To this end, a multiscale decomposition in space and time was performed to isolate the"mesoscale" wind work, characterized by a negative contribution to the total wind work (the so-called "eddy-killing" effect). The global average difference between ODYSEA estimates of wind work and COAS estimates at mesoscales is ∼0.06 mW/m$^2$, which is 1/10 smaller than typical values in energetic mesoscale regions. The estimates of wind work are insensitive to the uncorrelated noise between winds and currents. Finally, the comparison of ODYSEA-like sampling against synthetic observational datasets (global datasets like "AVISO", "QuikSCAT", and "GlobCurrent"), where the spatial and temporal resolutions of ocean currents and wind differ, further emphasize not only the importance of spatial collocation of wind stress and currents but also their contemporaneity on the integrated wind work.

One difference we found between the results of COAS and results the ODYSEA simulator is that wind work in the northern hemisphere (north of 40°N) in Jul-Aug-Sep differs from the wind work from the ODYSEA simulator, being ∼50% smaller. The only explanation we have so far is that this difference may be due to high-frequency low wind amplitudes during summer in this region that force near-inertial oscillations. Winds less than 5 m/s are not taken into account in the ODYSEA simulator, since the standard deviation of the surface current error becomes much larger than 1 m/s for these wind

amplitudes. One explanation for the underestimation is that even if the wind is low, its associated intermittency triggers the generation of near-inertial waves. Torres et al. [3] demonstrated that a large part of positive wind work is located at the near-inertial band. However, this contribution is masked out in summer at mid-latitudes. This does not occur in the summer southern hemisphere since wind magnitudes are usually larger than 5 m/s. This underestimation is not present the rest of the months, since the standard deviation of the wind speed is above 5 m/s, even in the northern hemisphere. This explanation needs to be confirmed.

**Author Contributions:** Conceptualization, H.T., A.W., E.R. and P.K.; methodology, H.T., A.W., T.L., J.W. and P.K.; software, H.T. and A.W.; A.W. lead the development of ODYSEA simulator; D.M. lead the development and implementation of the MITgcm-GEOS5 coupled ocean–atmosphere simulation (COAS); formal analysis, all authors contributed equally; writing—original draft preparation, H.T. and P.K.; writing—review and editing all authors contributed equally in this part. All authors have read and agreed to the published version of the manuscript.

**Funding:** This research was carried out in part at the Jet Propulsion Laboratory, California Institute of Technology, under a contract with the National Aeronautics and Space Administration (NASA) and funded through the internal Research and Technology Development program. High-End computing was provided by the NASA Advanced Supercomputing (NAS) Division at the Ames Research Center.

**Data Availability Statement:** ODYSEA simulator can be download from https://github.com/awineteer/odysea-science-simulator/ (accessed on 15 March 2023). The coupled ocean–atmosphere simulation can be found at: https://portal.nccs.nasa.gov/datashare/G5NR/DYAMONDv2/GEOS_6km_Atmosphere-MITgcm_4km_Ocean-Coupled/GEOSgcm_output/ (accessed on 15 March 2023). In particular, the dataset contained in the folder *geosgcm_surf* were used in this study. The variables used in this study are U (east-west velocity component),V (north-south velocity component), oceTAUX (east-west wind stress component), and oceTAUY (north-south wind stress component).

**Acknowledgments:** Copyright 2022 California Institute of Technology. US government sponsorship acknowledged. H.T., AW, T.L., J.W., E.R., D.M., H.Z., were supported by the NASA Physical Oceanography (PO) and Modeling, Analysis, and Prediction (MAP) programs. PK acknowledges support from the SWOT Science Team, the NASA S-Mode project and the QuickSCAT mission. High-end computing was provided by the NASA Advanced Supercomputing (NAS) Division at the Ames Research Center.

**Conflicts of Interest:** The authors declare no conflicts of interest.

**Appendix A**

The methodology to get access to the mesoscale wind work component (fully described in Torres et al. [3]) consists of decomposing wind stresses (or winds) and surface currents as follows:

$$\mathbf{X} = \overline{\mathbf{X}} + \mathbf{X}'_{hf} + \mathbf{X}'_{lf>} + \mathbf{X}'_{lf<} \, , \tag{A1}$$

where $\mathbf{X}$ represents either $\boldsymbol{\tau}$, $\mathbf{U_{10}}$, or $\mathbf{u_o}$. The overline operator represents a time average over 3 months, also called time-mean or seasonal-mean, and the prime operator represents time fluctuations with periods smaller than 3 months. The time fluctuations are further decomposed into a high-frequency component ($hf$) for periods smaller than 3 days and a low-frequency component ($lf$) for periods between 3 days and 3 months. The $hf$ component captures high-frequency contributions such as those at the inertial frequency. The low-frequency component is further decomposed into two contributions in terms of spatial scales: the large-scale contribution ($lf >$) for spatial scales larger than a critical length scale $L_c$ and the small-scale contribution ($lf <$) for scales smaller than $L_c$. Following Rai et al. [26], we define $L_c$ as the length scale for which the low-frequency component of wind work is negative for scales smaller than $L_c$ and positive for larger scales. Negative wind work at these scales has been referred to as "eddy killer" or "eddy damping" [6,9,22,26]. Using the same procedure as Rai et al. [26], we found that

$L_c \approx 250$ km (see Torres et al. [3] for more details). We apply Reynolds decomposition to Equation (2) using (3). The resulting wind work at each grid point averaged over 3 months includes a time-mean component ($\overline{\boldsymbol{\tau}}.\overline{\mathbf{u}_\mathbf{o}}$) and a total time-dependent component ($\overline{\boldsymbol{\tau}'.\mathbf{u_o}'} = \overline{\boldsymbol{\tau}'_{hf}.\mathbf{u_o}'_{hf}} + \overline{\boldsymbol{\tau}'_{lf>}.\mathbf{u_o}'_{lf>}} + \overline{\boldsymbol{\tau}'_{lf<}.\mathbf{u_o}'_{lf<}}$), such that

$$\overline{F_s} = \overline{\boldsymbol{\tau}}.\overline{\mathbf{u}_\mathbf{o}} + \overline{\boldsymbol{\tau}'_{hf}.\mathbf{u_o}'_{hf}} + \overline{\boldsymbol{\tau}'_{lf>}.\mathbf{u_o}'_{lf>}} + \overline{\boldsymbol{\tau}'_{lf<}.\mathbf{u_o}'_{lf<}} . \tag{A2}$$

The validity of the Reynolds decomposition was tested in Torres et al. [3]. The mesoscale wind work component corresponds to the last one in Equation (A2): $\overline{\boldsymbol{\tau}'_{lf<}.\mathbf{u_o}'_{lf<}}$.

**Appendix B**

The ocean-only simulation is a global ocean numerical simulation known as LLC2160 [29], which was carried out as a collaborative effort between the Massachusetts Institute of Technology (MIT) and NASA's Jet Propulsion Laboratory (JPL) and Ames Research Center (ARC). This simulation has the same configuration as the ocean component of COAS [3]. In addition, this simulation is forced by the six-hourly surface atmospheric fields (wind, air temperature, and humidity) from the 0.14° ( 15 km) European Center for Medium-Range Weather Forecasts (ECMWF) atmospheric operational model re-analysis.

The methodology to diagnose the mesoscale wind work component from LLC2160 surface currents and winds from ECMWF re-analysis is identical to the one used for COAS that is described in Section 2 and Appendix A. It requires determining a critical length scale $L_c$, following Rai et al. [26], to separate the large-scale contribution ($lf >$) (scales larger than $L_c$) and the small-scale contribution ($lf <$) (scales smaller than $L_c$) (see Rai et al. [26] and Torres et al. [3] for more details). Then, mesoscale wind work corresponds to $\overline{\boldsymbol{\tau}'_{lf<}.\mathbf{u_o}'_{lf<}}$ (Equation (A2)). An illustration of the methodology is given in Figure A1, which shows the mesoscale wind work as a function of the length scale, $L$, that separates low-frequency motions into large and small scales. Mesoscale wind work is negative and decreases from very small scales to $L_c$. For larger scales, the wind work increases. This means that wind work is negative for all scales smaller than $L_c$ and becomes positive for larger scales, hence the interpretation for $L_c$. Figure A1 further reveals that $L_c$ for LLC2160 is larger and concern mesoscale wind work with a larger magnitude than in COAS. This explains that the mesoscale wind work found in COAS is almost twice as small than in LLC2160. Note the $L_c$ value from LLC2160 and magnitude of the negative mesoscale wind work corresponding to this value are similar to those reported in Rai et al. [26].

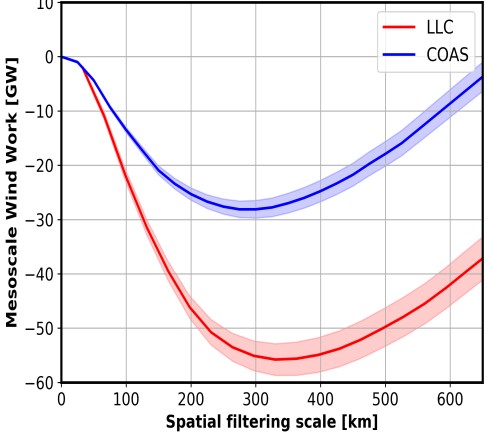

**Figure A1.** Low-frequency wind work from LLC2160 and COAS. $L_c$ is defined as the minimum of the blue (COAS) or red (from LLC2160) using 12-month outputs of wind stresses and ocean currents (see Rai et al. [26] and Torres et al. [3] for the calculation of these curves).

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
