# Peer review of "Anticipated Capabilities of the ODYSEA Wind and Current Mission Concept to Estimate Wind Work at the Air–Sea Interface"

_remotesensing, doi:10.3390/rs15133337_

Round 1

Reviewer 1 Report

Review of “Capabilities of the ODYSEA wind and current mission to estimate wind work at the air-sea interface” by Torres et al.

Overall comments:

This manuscript is well-written and rich in scientific virtues, specifically, the potential contributions for understanding the influences of mesoscale and early sub-mesoscale dynamics. 

The objects, background, methodology and data, and conclusions are nicely presented and explained, although some editorial changes and reorganizations of a few logical chains are needed.

Another question that I have is that throughout the paper, the bottom drag is not mentioned. I assume it’s lumped into the damping of mesoscale eddies, but I would like to ask the authors for the treatments on the bottom drag.

Editorial opinion: Minor revision

Please find the below detailed comments:

Location

Comments

Title 

The title seems ambiguous. It does not clearly mention the study is based on a simulator and a conceptual mission.

L22-28

The partitions of the wind work into a single-digit percentage precision (28% and 44%) sound odd. Unless it is unconditional, I suggest adding a short phrase with a source of these numbers. 

L43-44

Does the SWOT launched last December satisfy these conditions?

L107-109

When describing the spatial resolutions of the grid spacing, suggest a brief explanation of the corresponding dynamic motions to these scales (e.g., refer to Appendix B). 

L114-119

(also following paragraphs) These paragraphs are not quite coherent and hard to read. One reason is that the explanation of the figure starts before raising the question that this figure can answer. 

For example, you may move the last sentence on L119 to L114 as the first sentence.

L123 and L140

On L140, “this signal” is ambiguous, but it is also less clear what ODYSEA measures (wind and current)

L184-204

These two paragraphs are hard to read. Suggest emphasizing the questions or findings at the beginning of the paragraph and then expanding with the details.

L219

(and other locations) It may be better to add a reference (e.g., the Coriolis frequency f or 2f) when describing the high and low frequencies.

L242

This paper also discusses a similar idea for the damping of mesoscale eddies, which can be added to the citations.

Taylor and Straub (JPO 2020) DOI: 10.1175/JPO-D-19-0299.1

Also, how about the Ekman effects? For sure the Ekman effects are much faster and finer in spatial scale, but it cannot be inferred from the local velocity or wind stress either.

Wenegrat and Thomas (JPO 2017) DOI: 10.1175/JPO-D-16-0239.1

L242 L310

(multiple locations) “X is times smaller than Y” does not sound like a formal and correct expression. Probably you mean “1/X smaller than”.

L319

This paragraph is mostly repeating the paragraph in L222.

Author Response

We thank the two reviewers for constructive comments and suggestions, which we have considered in the revised version of the manuscript, as summarized below.

This manuscript is well-written and rich in scientific virtues, specifically, the potential contributions for understanding the influences of mesoscale and early sub-mesoscale dynamics.

The objects, background, methodology and data, and conclusions are nicely presented and explained, although some editorial changes and reorganizations of a few logical chains are needed.

Editorial opinion: Minor revision

Another question that I have is that throughout the paper, the bottom drag is not mentioned. I assume it’s lumped into the damping of mesoscale eddies, but I would like to ask the authors for the treatments on the bottom drag.

            Thank you for the comment. In the revised version of the manuscript, we have added the next statement (L114):

“The numerical simulation cannot resolve the mechanical boundary layers that must be present at the air-sea interface and seafloor, resulting in the transfer of momentum. The numerical machinery uses a drag-stress at the ocean top and bottom that scales quadratically with the near-boundary velocity [16]”.

The title seems ambiguous. It does not clearly mention the study is based on a simulator and a conceptual mission.

            Thank you for the comment. In the present revised version, we have changed the title: “Anticipated capabilities of the ODYSEA wind and current mission concept to estimate wind work at the air-sea interface”.

L22-28 The partitions of the wind work into a single-digit percentage precision (28% and 44%) sound odd. Unless it is unconditional, I suggest adding a short phrase with a source of these numbers.

            Thank you for the comment. In the revised version we have made the corrections in the main text (L22-29):

“Approximately ∼ 28% of this wind work is used to generate near-inertial oscillations and internal gravity waves (with spatial scales of 20 to 1000 km and time scales of hours), which impact ocean mixing and therefore contribute to setting global stratification and large-scale circulation [3– 5]. Another ∼28% contributes to force and damp mid-latitude currents, such as those associated with mesoscale eddies (with spatial scales from 50 km to more than 500 km and time scales of days to months), which are critical players in the horizontal and vertical transport of heat at these latitudes [3 ,6, 7]. The remaining ∼44% forces large-scale zonal jets with time scales of a few months,…”

L43-44 Does the SWOT launched last December satisfy these conditions?

            Thank you for the question. Unfortunately, SWOT measurements will have a revisit time period of 11 days and 1-day during the fast-sampling phase but only for 6 months.

L107-109 When describing the spatial resolutions of the grid spacing, suggest a brief explanation of the corresponding dynamic motions to these scales (e.g., refer to Appendix B).

            Thank you for the comment. In the revised version of the manuscript, we have added the next information (L109):

            “In terms of physical scales, the spacing grid resolution of ~ 6km and ~ 4km are sufficient to resolve length scales between 25-km and 30-km (Soufflet et al., 2016).”

L114-119 (also following paragraphs) These paragraphs are not quite coherent and hard to read. One reason is that the explanation of the figure starts before raising the question that this figure can answer.  For example, you may move the last sentence on L119 to L114 as the first sentence.

Thank you for the comment. In the revised version of the manuscript, we have made the correction (L119-120):

“Surface currents involve much smaller scales than winds and evolve more slowly. This is illustrated in Fig.1 that shows surface currents (left panel) and winds (right panel) sampled by the ODYSEA simulator in the global ocean. The wind field involves large scales O(1000km), resulting from atmospheric weather patterns that propagate rapidly. Embedded within these large-scale patterns are smaller-scale patterns (as small as 100 km), some of them propagating with the large-scale ones. Energetic surface currents include near-inertial motions and internal tides, mesoscale eddies, and zonal jets.”

L123-140 On L140, “this signal” is ambiguous, but it is also less clear what ODYSEA measures (wind and current)

            Thank you for the comment. In the revised version of the manuscript, we have specified that we refer to the signal-to-noise ratio (L146):

            “…in which the signal-to-noise ratio is dominated by the random noise…”

L184-204 These two paragraphs are hard to read. Suggest emphasizing the questions or findings at the beginning of the paragraph and then expanding with the details.

            Thank you for the comment. In the revised version of the manuscript, we have added the sentence at the beginning of the paragraph (L190).

“Global estimates of the total wind work using ODYSEA-N is consistent with COAS.”

L219 (and other locations) It may be better to add a reference (e.g., the Coriolis frequency f or 2f) when describing the high and low frequencies.

            Thank you for the comment. In the revised version of the manuscript, we have specified that high-frequency motions are associated with frequencies > 1/3-day and low-frequency motions are associated motions are associated with frequencies < 1/3-day.

L242 This paper also discusses a similar idea for the damping of mesoscale eddies, which can be added to the citations. Taylor and Straub (JPO 2020) DOI: 10.1175/JPO-D-19-0299.1

            Thank you for the comment. In the revised version of the manuscript, we have added the next paragraph (L248-251):

“This wind work is mostly negative, corresponding to the damping of mesoscale eddies, and has a net magnitude 10–20 times smaller than the total wind work. The physics of this damping of mesoscale eddies is discussed in [3] (see also [18]).”

Also, how about the Ekman effects? For sure the Ekman effects are much faster and finer in spatial scale, but it cannot be inferred from the local velocity or wind stress either.  Wenegrat and Thomas (JPO 2017) DOI: 10.1175/JPO-D-16-0239.1

            Thank you for the comment. In the revised version of the manuscript, we have added the next sentence (L252):

“Note that this physics includes the Ekman pumping [18]) since the latter is implicitly included in the full momentum equations resolved by the coupled model.”

L242-310 (multiple locations) “X is times smaller than Y” does not sound like a formal and correct expression. Probably you mean “1/X smaller than”.

            Thank you for the comment. In the revised version of manuscript, we have corrected the sentences (L247-326).

L319 This paragraph is mostly repeating the paragraph in L222.

            Thank you for the comment. In the revised version of the manuscript, we have improved the paragraph (L328).

“One difference we found between COAS and results from the ODYSEA simulator is that wind work differs in the northern hemisphere (north of 40◦N) in Jul-Aug-Mar with the wind work from the ODYSEA simulator being ∼50% smaller. The only explanation we have so far is that this difference may be due to high-frequency low wind amplitudes during summer in this region that force near-inertial oscillations. Winds less than 5 m/s are not taken into account in the ODYSEA simulator since the standard deviation of the surface current error becomes much larger than 1 m/s for these wind amplitudes. One explanation for the underestimation is that even if the wind is low, its associated intermittency triggers the generation of near-inertial waves. Torres et al. [3] demonstrated that a large part of positive wind work is located at the near-inertial band. However, this contribution is masked out in summer at mid-latitudes. This does not occur in the summer southern hemisphere since wind magnitudes are usually larger than 5 m/s. This underestimation is not present the rest of the months, since the standard deviation of the wind speed is above 5 m/s, even in the northern hemisphere. This explanation needs to be confirmed.”

Reviewer 2 Report

The purpose of this study is to evaluate the capabilities of ODYSEA. Using a measurement simulator fed by a coupled ocean-atmosphere model, the study estimates wind work on a global scale. The results indicate that the ODYSEA instrument performs well globally, with the exception of some latitudes. This performance can be attributed to ODYSEA's wide-swath properties. Additionally, the instrument provides twice-a-day coverage at mid-latitudes (low latitudes) and demonstrates insensitivity to uncorrelated errors in the estimated wind and current, contributing to its overall effectiveness. I consider that the subject addressed in this study is worthy of investigation and it is appropriate for the Journal of Remote Sensing. Also, I consider that the manuscript is almost ready to be published in the journal, it needs a minor review. Below, I have attached a few points and details that should be reviewed before publication.

p { margin-bottom: 0.1in; direction: ltr; color: #000000; line-height: 115%; text-align: left; orphans: 2; widows: 2; background: transparent }p.western { font-family: "Liberation Serif", serif; font-size: 12pt; so-language: en-US }p.cjk { font-family: "Noto Serif CJK SC"; font-size: 12pt; so-language: zh-CN }p.ctl { font-family: "Lohit Devanagari"; font-size: 12pt; so-language: hi-IN }

Author Response

We thank the two reviewers for constructive comments and suggestions, which we have considered in the revised version of the manuscript, as summarized below.

Referee #1

I consider that the subject addressed in this study is worthy of investigation and it is appropriate for the Journal of Remote Sensing. Also, I consider that the manuscript is almost ready to be published in the journal, it needs a minor review.

Line 60: The period is missing at the end of the sentence.

            Thank you for the comment. In the present revised version, we have added the period at the end of the sentence.

Line 69: Which means ESE?, is not previously defined.

            Thank you for the comment. In the present revised version, we have added the definition of ESE in the line 69: “NASA Earth System Explorer”.

Figure 3: “smaller than 5 m/s; The right” change “;” with a period.

            Thank you for the comment. In the present revised version, we have changed “;” with a period in Figure 3.

Lines 125-126 and 130: The authors could please justify why 10% of the wind and why a threshold of 10 and 1 m/s and 5 m/s, respectively.

            Thank you for the comment. In the present revised version, we have included more information about the thresholds (L128-137).

“The ODYSEA simulator estimates wind and current errors and adds them onto collocated winds and currents from COAS. The simulator uses a simple model for the wind errors to simulate ODYSEA performance requirement: the error standard deviation corresponds to 10% of the wind speed when the speed is larger than 10 m/s and 1 m/s for smaller wind speeds. For surface currents, an error model based on [12] was used to form a lookup table based on wind speed, wind direction, and swath position. At the center and edges of the swath, the resulting surface current errors are large. In this study, we exclude the center 100km and 50km on either edge of the swath, in addition to areas with winds less than 5m/s because the standard deviation of the error is larger than the science requirements.”

Figure 7: The legend in the left and right panels appears to be inconsistent, as it shows a value of 1 and 1.0 m/s, respectively.

Thank you for the comment. In the present revised version, we have updated Figure 7 according to reviewer’s comment.

Line 282: A space is needed “… from Rai …”

Thank you for the comment. In the present revised version, we corrected the typo.

The authors could please provide a little more discussion regarding the underlying physics that contribute to the increase in error at different latitudes compared to the observations.

Thank you for the comment. In the present revised version, we have added the sentence in Line 328:

 “Winds less than 5 m/s are
not taken into account in the ODYSEA simulator since the standard deviation of the surface
current error becomes much larger than 1 m/s for these wind amplitudes. One explanation
for the underestimation is that even if the wind is low, its associated intermittency
triggers the generation of near-inertial waves. Torres et al. [3] demonstrated that a large
part of positive wind work is located at the near-inertial band. However, this contribution is masked out in summer at mid-latitudes.”

Line 302, 305, 310, 323, and 325: The units are misspelled, according to the rest of the manuscript.

Thank you for the comment. In the present revised version, we corrected the typo.

The authors could briefly describe or explain what happens the rest of the months.

Thank you for the comment. In the present revised version, we have added the next paragraph in Line 339: “This underestimation is not present the rest of the months, since the standard deviation of the wind speed is above 5 m/s, even in the northern hemisphere.”
